# Research on the Input–Output Model of the Rural Agricultural Eco-Economic System Based on Emergy Theory

Yulin Zhu [1,2,*], Mingjie Li [1,3,*], Shiwei Lu [1], Huimin Wang [1], Jinjin Wang [1] and Wei Wang [1]

1   College of Economics, Central South University of Forestry and Technology, Changsha 410004, China;
    lushiwei_123@163.com (S.L.); wanghuimin2022@126.com (H.W.); wangjinjin2020@126.com (J.W.);
    wangwei11132@126.com (W.W.)
2   Hunan Green Development Research Institute, Changsha 410004, China
3   School of Finance, Guangzhou Huashang College, Guangzhou 511300, China
*   Correspondence: t19970886@csuft.edu.cn (Y.Z.); limingjieping@163.com (M.L.)

**Abstract:** Based on the Wassily W. Leontief input–output model, this paper constructs an emergy input–output model of an agricultural eco-economic system based on emergy theory. This model overcomes three limitations of the traditional input–output model. First, the conversion rate of ecological energy is used to solve the problem posed by the fact that the material input–output table cannot be directly combined due to the different measurement units of different substances. Second, it transforms the research object from a pure economic system to an eco-economic system by adding resources and environmental factors. Third, this paper solves the problem posed by the fact that the value input–output model is greatly affected by inflation, and uses the model to analyze the input–output of Zoumajie town in Shuangfeng City, Hunan Province, China, by unifying dimension (solar energy value).

**Keywords:** emergy theory; input–output model; agro-ecosystem; Zoumajie town

## 1. Introduction

Wassily W. Leontief developed the input–output analysis model of the economic system in 1931. He compiled the input–output tables of the United States in 1919 and 1921 using census data on the national conditions to analyze the economic structure and economic balance [1]. This method is called the department connection method in the Soviet Union and the industry connection method in Japan. It is a quantitative analysis method for studying the interdependence of inputs and outputs among various sectors of the socio-economic system [2]. However, with the rapid development of the economy, resource and environmental problems have become the focus of attention, and inflation caused by the issuance of additional money has occurred from time to time [3,4].

Three application limitations of the traditional model have become increasingly obvious: first, the traditional model can be divided into two types: a material type and a value type. The material input–output model takes the material quantity unit of the product as the unit of measurement, which can better reflect the quantitative dependence of the material product in the production process. However, due to the different units of various substances and resources in the materialmaterial input–output type, the table cannot be directly combined. Second, the value input–output model takes currency as the unit of measurement, however, the value of currency is based on the socially necessary labor time. It measures only the labor amount paid by human beings in the process of resource utilization and ignores the value and contribution of natural resources and the environment. In other words, the object of analysis of the traditional input–output technique is limited only to the economic system, rather than the eco-economic system, which includes resources and the environment. Third, the value input–output model is easily affected by price factors.

In recent years, scholars have performed much research on improving the input–output models. Matteo V. Rocco et al. (2016) added energy factors as endogenous variables into the model and applied the model to analyze the energy consumption level of national products. Kander A et al. (2015) introduced environmental factors into the input industry model and applied the model to analyze the division mechanism of carbon emission reduction responsibility. Lin J et al. (2014) applied the model to analyze the causes of air pollution. Zhao H Y et al. (2015) applied the model to analyze the implied emissions from regional trade. The following scholars have tried to apply emergy theory to the input–output model: Cheol-joo Cho (2013) conducted a preliminary study on the solution to the energy conversion coefficient matrix. Baral. A and Bakshi.B.R. (2010) used the mixed-energy input–output model to analyze the American economy and proved that this model could replace the simple input–output model. Huibin Du et al. (2011) analyzed the driving factors of $CO_2$ emissions in Sino-US trade using the energy input–output model. L.X. Zhang et al. (2017) used an emergy input–output model to analyze resource consumption among various sectors in China. S.Y. Zhou et al. (2010) set up a systematic energy input–output model and performed a quantitative analysis of Beijing's urban economy and ecological environment.

In China, Feng Tong, Zhao Hongyan et al. (2016) used input–output models to calculate greenhouse gas emissions. Zhong Yongde, Shi Chenyi et al. (2015) applied the model to study the regional carbon emission level. Tang Decai, Xue Peipei et al. (2016) applied the model to study the efficiency of environmental governance. Studies introducing energy consumption factors into the input–output model include the following. Wang Shijin (2015) used the model to calculate and study the implied energy consumption of Jiangsu's foreign trade. Liu Liqiu, Ma Jiajun et al. (2015) calculated and studied the total energy consumption level of China's construction industry. Liao Mingqiu (2011) compiled an input–output model of "energy conservation and emission reduction" by combining energy, the environment and the economy. Few studies have introduced emergy theory into the input–output model. Only Liu Yifang and Tong Rencheng (2011) have theoretically discussed the model of the circular economy.

These research results have played a positive role in overcoming the application limitations of the input–output model, however, most studies on model optimization or improvement can only avoid one or two of the three limitations mentioned above. For example, the studies of Matteo V. Rocco et al., Kander A et al., Lin J et al., Zhao H Y et al., Feng Tong et al., Zhong Yongde et al., Tang Decai et al., Wang Shijin et al., and Liao Mingqiu could overcome only the first two of the three limitations. Although Huibin Du et al., L.X. Zhang et al., and S.Y. Zhou et al. introduced emergy theory, only one of the two factors of resource consumption and environmental pollution was introduced into the input–output model. Only Liu Yifang and Tong Rencheng (2011) tried to introduce emergy theory into the input–output model, however, their research focused on the circular economic system rather than the eco-economic system in the ordinary sense, and they conducted only a theoretical exploration. The results of the research by Cheol-joo Cho, Baral. A and Bakshi. B.r. provide us with new ideas and enhance our confidence in model improvement.

Meanwhile, based on the principles of system ecology, energy ecology and eco-economics, American ecologist Professor H. T. Odum developed emergy analysis theory, which is a theory that provides a new idea for the study of input-output. The advantage of this theory is that the ecological factors (resources and the environment elements) and economic factors are combined effectively through a unified dimension (ecological), which can effectively compensate for the defect of the fact that it is hard for the non-marketing input to use monetary units. Additionally, the theory can avoid the influence of inflation on the results of the analysis, indirectly improving the precision of prediction and assessment. LAN Shengfang (2002) believes that emergy theory is the most advanced theory of sustainable development in current ecosystem theory. Qi Hongqian Ye (2016) believes that the solution to input–output and conversion coefficients based on ecological energy conversion is one of the frontiers and development trends of current eco-economic research.

Accordingly, this paper uses emergy theory in eco-economics to improve and optimize the input–output model. Through the conversion, all units of input–output elements can be converted into emergy, and the monetary unit is no longer used as the measurement unit. These features can not only improve the applicable scope and estimation accuracy of the input–output analysis technique, but also comprehensively overcome the three limitations of the traditional model. First, the conversion rate of ecological energy is used to solve the problem posed by the fact that the material input–output table cannot be directly combined due to the different measurement units of different substances. Second, the model transforms the object of research from a pure economic system to an eco-economic system by adding resources and environmental factors. Third, this paper solves the problems posed by the fact that the value model is greatly affected by inflation. Moreover, this paper takes the agricultural ecosystem of Zoumajie town in Shuangfeng City, Hunan Province, China, as the research object and empirically studies the input structure and production mode of the ecosystem elements, providing an empirical reference for the optimization of the rural industrial structure and sustainable development.

## 2. Research Model

### 2.1. Input–Output Model and Emergy Theory

Input–output analysis is a method for researching the equilibrium of relationships between each department of the national economy [5,6]. Based on the hypothesis of general equilibrium, it shows the dependence of the product volume of each economic sector in the form of a system of equations. Then, based on the statistical material of the data, a kind of equilibrium table can be made with the form of a matrix or chessboard to show an overall perspective on the equilibrium between the supplement and demand of products in each department and to obtain the ratio of each department's product quantity and other departments' product quantity needed to meet this department's product quantity [7,8]. The first step is to make an input–output table. Based on the production of different products, this table divides the entire economic system into different sectors, which shows the economic system's overall perspective on industry relationships. The table contains column and row items. The columns in the table show each department's input and its composition, while the rows show the products' output and to where they are distributed [9].

In the input-output table, inputs are subdivided into intermediate inputs and initial inputs. Intermediate inputs refer to various raw materials and fuels put into the production process, namely various material consumption. While initial inputs refer to the expenditure of wages, rent and interest required in the production process. Outputs are divided into intermediate and final products. Intermediate and final products are distinguished by whether the products are returned to the production process during the period under review. Based on the finished input–output table, the relationship between columns and rows is analyzed to infer the influence on other departments if a department's input and output changed and to calculate the total amounts of all kinds of demanded products to satisfy the social "final consumption" and to finally predict the economic development future of the research object.

The emergy analysis theory developed by Professor H. T. Odum measures all kinds of resources, products, labor, and information in the ecological system and eco-economics system. These elements have different natures and are incomparable in the form of unified solar emergy, solar emergy (sej) is served as a unified unit of measurement. Thus, emergy analysis theory is used to objectively evaluate and compare the contributions made to the human economic system by various natural resources and to evaluate the sustainable development capacity of all kinds of ecological systems based on it [10,11].

Traditional physical analysis methods cannot reflect the relationship between the various elements and environment of the agricultural ecosystem. They can only reflect the economic relationship between the various elements of the agricultural ecosystem. In the meantime, due to the problems of measurement units between the input and output of elements, the elements cannot be merged directly in the table. Thus, the traditional

physical agricultural input–output model can satisfy only the transverse balance, not the longitudinal balance [12]. As a result of this great limitation, we try to combine emergy theory and the input–output model. Based on the physical type of the input–output table and resource and environmental factors taken as new endogenous variables, the input–output table (tentatively named the resource-environment-economic input–output table) based on the ecological energy conversion of the eco-economic system is compiled and modelled in this paper. The basic idea of this paper is as follows:

(1) Constructing the resource-environment-economic input–output table;

① The first quadrant of this table is exactly the same as the traditional GDP input–output table. It is the intermediate input (intermediate use) quadrant, reflecting the consumption of various products (including goods and services) in the production process of a particular ecosystem.

② The second quadrant of the table adds resource expenditures and environmental outputs to the original input–output table to reflect the end use of products.

③ The third quadrant of this table is resource consumption and environmental pollution, reflecting the resource consumption and environmental pollution caused by the production process of an ecosystem.

④ The fourth quadrant of this table is the environmental pollution quadrant in the consumption domain, which considers the virtual control cost of pollution caused by the consumption of an ecosystem. It reflects the one-to-one corresponding relationship between resource consumption in the production process and environmental pollution as well as resource and environmental expenditure in the final use of the consumption domain.

⑤ The fifth region of the table is the green GDP quadrant, which reflects the amount of green GDP generated in the production process of a particular region.

(2) Data Collecting and model construction. We need to collect and organize data, compile resource-environment-economic input–output tables and build input–output analysis models;

(3) Energy conversion. We carry out special research and accounting and comply with the energy conversion coefficient and energy conversion coefficient matrix of each material resource in the resource-environment-economic input–output table;

(4) Matrix solution. We need to solve the matrix of the resource-environment-economy input–output table of the eco-economic system, including calculating the direct and complete consumption coefficient of natural resources, the direct pollution emission index and the complete pollution emission index of eco-economic system operation;

(5) Research and application. Based on the solution results, we analyze the impact of the economic and social activities of the eco-economic system on resource consumption and environmental pollution, carry out prediction research on resource consumption and environmental pollution emissions of the eco-economic system, and propose corresponding countermeasures and suggestions.

*2.2. Model Construction*

2.2.1. Draw the Energy System Diagram

In general, agro-ecosystems consist of four subsystems: farming, animal husbandry, forestry and fisheries. The energy input from the external systems include three categories: (1) Environmental resource energy. It can be invested in the agricultural ecosystem mainly in the form of sun and rain; (2) Industrial auxiliary energy. It includes non-renewable energy such as fertilizer, pesticide, agricultural film, agricultural machinery, agricultural electricity, agricultural diesel and other non-renewable resources and energy; (3) Organic energy. It includes energy such as labor, animal power, and organic fertilizer [13]. The system is dominated by crops, animal husbandry, forestry, fisheries and agricultural sideline exports. Based on the knowledge of the ecological environment and social economy, the energy system diagram of agro-ecological economic system is drawn using H.T. Odum's 'Energy System Language' legend, which lays the foundation for compiling the energy

input–output table of the agro-ecological economic system. The energy system diagram of the agro-ecological economic system is shown in Figure 1 and Table 1.

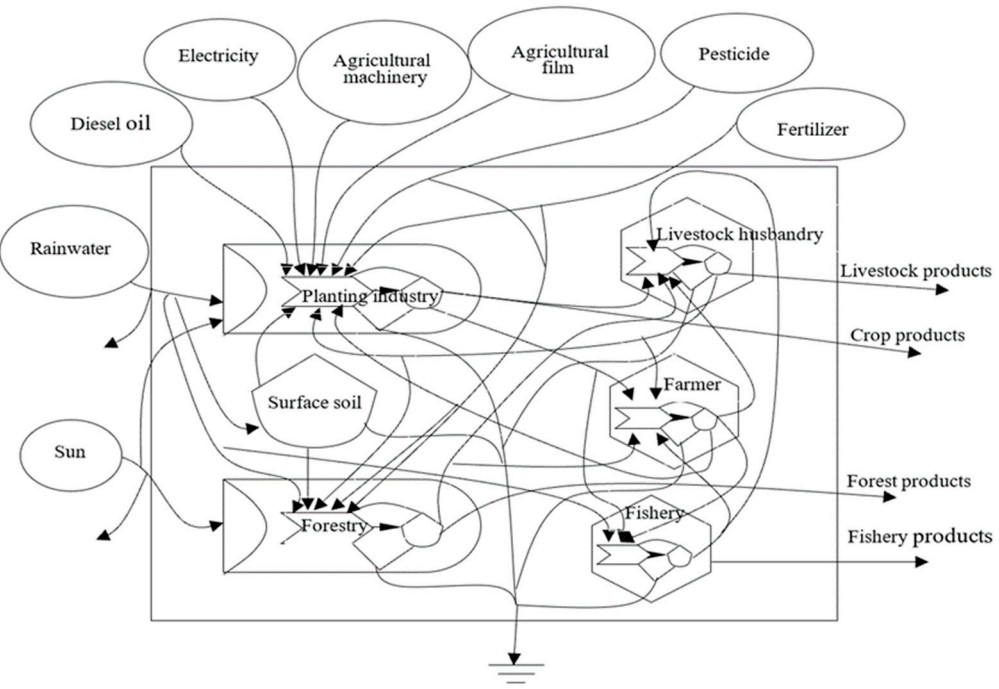

**Figure 1.** Energy system diagram of the agricultural eco-economic system.

**Table 1.** Main symbols of energy systems and their usage.

| Symbol | Component Name | Explanation |
|---|---|---|
| | Energy sources | Represents all forms of energy (material) input from outside of the system |
| | System border | The rectangular box used to represent the boundary of the system is the dividing line between the inside and outside of the system |
| | Storehouse | The place where energy is stored in the system, such as soil and groundwater |
| | Heat storage tank | The dissipation of energy; the energy to be released from storage, work, and components |
| | Producer | Biological producers such as plants |
| | Consumer | Refers to microorganisms, animals, etc., usually heterotrophic organisms |

### 2.2.2. Economic Department Setting

Based on the energy system diagram of the agricultural eco-economic system drawn in the above section, according to the principle of inter-departmental contact and eco-economic benefits, the specific settings of the departments in the process of compiling the emergy input–output table of the agricultural eco-economic system areas are as follows:

(1) Planting industry. The products of this department mainly include grains, beans, potatoes, cotton, oilseeds, vegetables, tobacco, Chinese herbal medicine, tea and fruits;

(2) Livestock husbandry. The products of this department mainly include pork, beef, lamb, poultry, rabbit, honey and eggs;

(3)　Forestry. The products of this department specifically include camellia seeds, dried bamboo shoots, nuts, wood and bamboo;

(4)　Fishery. The products of this department are mainly freshwater aquaculture products;

(5)　Agricultural sidelines. The main product of this department is straw;

(6)　Industrial auxiliary energy input department. The products provided by this department to the agricultural eco-economic system mainly include chemical fertilizer, pesticide, agricultural film, agricultural machinery, agricultural diesel oil, agricultural electricity and concentrated feed;

(7)　Organic energy input department. The products provided by this department to the agricultural eco-economic system mainly include labour, animal power and organic fertilizer.

It should be explained that the grain products of the planting industry include rice, corn and sorghum. Straw is an agricultural side-line product. When calculating the agricultural input of crops such as grains, the energy consumed by planting straw is also included. To avoid repeating the input, when processing the straw data, this paper assumes that straw has only input behaviour for agricultural production activities and does not consume the products from other departments. Although livestock such as pigs, cattle and sheep are in this area, the products of animal husbandry are produced only in the form of products such as pork, beef and mutton. The output of products such as pigskin, cowhide and wool is not included. The area does not plant grass to feed livestock. It mainly uses roadside weeds for feeding and production and includes them in the agricultural ecosystem only in the form of green fodder. Therefore, when setting up the department, grass should be classified as an organic energy input. In addition, to pursue rapid production, livestock are fed concentrated feed. To clarify the corresponding relationship of the input–output table and to ensure that the ingredients of the concentrate contain a certain proportion of hormones, the concentrate is added to the column of the industrial auxiliary energy input.

2.2.3. Compilation of the Emergy Input–Output Table

Based on emergy theory, the emergy input–output table of the agricultural eco-economic system is compiled based on the material input–output table and the situation of agricultural production. The table can reflect not only the economic relations among the products but also the ecological connection between the products in the region. The study provides a new research method for the sustainable development of the agricultural eco-economic system in this area. The specific content of the emergy input–output table of the agricultural eco-economic system is shown in Table 2.

The row items of the table reflect the use of agro-economic system products based on economic purposes. In the traditional material input–output table, the row items are composed of intermediate products and final products. The intermediate products are the products that still need to be processed in the production field during the period. The final products are the products that have been processed in the production field in this period, which can be used. The sum of intermediate and final products is the number of total products.

In the table, agricultural products can be divided into non-commodity products and commodity products. Non-commodity products are used by farmers for daily consumption and agricultural production and consumption, such as the agricultural products consumed by farmers themselves. Commercial products are those that farmers sell to the rest of the economy.

**Table 2.** Emergy input–output table of the agro-ecological economic system.

| Input | Output | Intermediate Use | | | | | Finally, Using | | | | Total | Balanced Differences | Total Output |
|---|---|---|---|---|---|---|---|---|---|---|---|---|---|
| | | | | | | | Agricultural Self-Retention | | Commodity Products | | | | |
| | | Planting Industry | Forestry | Animal Husbandry | Fishery | Agricultural and Sideline | Self-Sufficiency Consumption | Storage | Non-Agricultural Intermediate Products | Final Agricultural Products | Total | Balanced Differences | Total Output |
| Intermediate input | Planting industry | x11 | x12 | x13 | x14 | x15 | c1 | s1 | i1 | f1 | y1 | e1 | O1 |
| | Forestry | x21 | x22 | x23 | x24 | x25 | c2 | s2 | i2 | f2 | y2 | e2 | O2 |
| | Animal husbandry | x31 | x32 | x33 | x34 | x35 | c3 | s3 | i3 | f3 | y3 | e3 | O3 |
| | Fishery | x41 | x42 | x43 | x44 | x45 | c4 | s4 | i4 | f4 | y4 | e4 | O4 |
| | Agricultural and sideline | x51 | x52 | x53 | x54 | x55 | c5 | s5 | i5 | f5 | y5 | e5 | O5 |
| Initial input | Industrial auxiliary energy | a1 | a2 | a3 | a4 | a5 | | | | | | | |
| | Organic energy | g1 | g2 | g3 | g4 | g5 | | | | | | | |
| Total input | | I1 | I2 | I3 | I4 | I5 | | | | | | | |

In the table, the row items are composed of four quadrants: The first quadrant is agricultural intermediate products, i.e., the amount of agricultural products consumed in the process of agricultural production. The second is the products retained by rural areas, i.e., the products that farmers use for their own consumption and storage. The third is non-agricultural intermediate products, which refer to the agricultural products that are sold to food processing plants, leather factories or textile factories for reprocessing when agricultural products are sold. The fourth is commodity products, which refer to those products that are directly sold in rural areas and those that are consumed by final consumers in urban areas.

The column items of the table shows the agricultural products, industrial auxiliary energy and organic energy consumed by each department in the production process. The column items of the table consists of only two quadrants: The first reflects the agricultural product input required in the production process of agricultural products, i.e., intermediate input. The fifth reflects the industrial auxiliary energy input and organic energy input required in the production process of agricultural products, i.e., initial input.

It needs to be explained that the first quadrant is the area corresponding to the intermediate input and intermediate use. The second is the area corresponding to the intermediate input and agricultural self-retention. The quadrant is the area where the intermediate input and non-agricultural intermediate products correspond. The quadrant is the area where the intermediate input corresponds to the final agricultural products. The fifth is the area corresponding to initial input and intermediate use.

### 2.2.4. The Equilibrium Relationship between the Rows and Columns

The equilibrium relationships of the emergy input–output table of the agro-ecological economic system are as follows:

Total output emergy = intermediate products use emergy + rural retained emergy + non-agricultural intermediate product emergy + final product emergy.

Total input emergy = intermediate products input emergy + industrial auxiliary energy input emergy + organic energy input emergy.

The row relationship is as following:

$$\begin{cases} (x_{11} + x_{12} + \ldots + x_{15}) + (c_1 + s_1) + (i_1 + f_1) = x_1 \\ (x_{21} + x_{22} + \ldots + x_{25}) + (c_2 + s_2) + (i_2 + f_2) = x_2 \\ \quad \ldots \\ (x_{51} + x_{52} + \ldots + x_{55}) + (c_5 + s_5) + (i_5 + f_5) = x_5 \end{cases}$$

The column relation is as follows:

$$\begin{cases} (x_{11} + x_{21} + \ldots + x_{51}) + (h_1 + g_1) = x_1 \\ (x_{12} + x_{22} + \ldots + x_{52}) + (h_2 + g_2) = x_2 \\ \quad \ldots \\ (x_{15} + x_{25} + \ldots + x_{55}) + (h_5 + g_5) = x_5 \end{cases}$$

Since the emergy input–output model of the agro-ecological economic system is based on the material input–output model, emergy is used to transform the data so that the measurement method is unified. Therefore, the relationship expression of the row and column balance can be obtained, however, the total output and total input data do not equally correspond.

### 2.2.5. Calculation of the Direct Consumption Coefficient

The direct emergy consumption coefficient $\partial_{ij}$ is the direct consumption of the products of sector *I* in the unit output of sector *J*. Its expression is as following:

$$\partial_{ij} = \frac{x_{ij}}{x_j}$$

In the above formula, the numerator $x_{ij}$ represents the value of product $i$ consumed in the production of department $j$, and the denominator $x_j$ is the column summation of the input–output table, which is the total input of department $j$. Since the input of any department is equal to the output of that department, $\partial_{ij}$ means the value of product $i$ consumed in each unit of output of department $j$.

2.2.6. Calculation of the Complete Consumption Coefficient

The direct consumption coefficient represents the direct consumption of products from department $I$ by unit output of department $J$, however, the indirect consumption from department $J$ to department $I$ is not included. Complete consumption is the sum of direct economic consumption and indirect consumption.

The calculation and derivation process of the complete consumption coefficient matrix is shown as follows:

$$b_{ij} = \partial_{ij} + b_{i1}\partial_{1j} + b_{i2}\partial_{2j} + \cdots + b_{ii}\partial_{ij} + \cdots b_{in}\partial_{nj}$$

If the above expression is expressed in matrix form, then

$$B = A + BA$$

In the formula, $A$ is the direct consumption coefficient matrix. $B$ is a complete consumption coefficient matrix, which is a matrix of $n$ including $n \times n$ complete consumption coefficients.

A further derivation can be made from the above equation: As $B(I - A) = A$, there is $B = A(I - A)^{-1}$.

This indicates that the direct consumption coefficient matrix is multiplied by a Leontief inverse matrix on the right to obtain the complete consumption coefficient matrix:

$$B = [I - (I - A)](I - A)^{-1} = (I - A)^{-1} - I$$

Therefore,

$$B = (I - A)^{-1} - I$$

**3. Empirical Analysis**

*3.1. Research Object*

Zoumajie town is located in north-western Shuangfeng City, Hunan Province, China, situated between 112°2′10″–112°11′15″ E longitude and 27°2′10″–27°35′50″ N latitude. It covers an area of 107.8 km$^2$, and it includes 41 villages, one neighbourhood committee, and a population of 74,000. Its agricultural population is 39,000. There are 3790.5 hectares of cultivated land, including 2992.2 hectares of paddy fields.

It is only 10 kms away from Shuangfeng City and 30 kms away from Loudi. Provincial Highway 210 from north to south and the Tan-Shao Expressway from east to west pass through the town centre. The traffic location advantage is very obvious. The town belongs to the one-hour economic circle of Changsha, Zhuzhou, Xiangtan, Shaoyang, and Hengyang. It is a pilot town for the reform of urban-rural integration, and it is one of the key towns in the construction of small towns in Loudi city.

Recently, Zoumajie town has experienced a major agricultural industrial structure adjustment. While stabilizing food production, the town is focusing on cultivating high-quality and high-efficiency agricultural production bases, and it has embarked on the path of industrial development. It has also constructed a high-standard planning modern technology demonstration park in Shuangfeng City, finishing works such as institution setting and functional specifications. With a solid agricultural foundation and preferential investment promotion policies, the construction of the core area has begun to take shape, and the prospects are very attractive. Thus, the strategic adjustment of the agricultural industrial structure of the town will be on a healthy and orderly development track.

*3.2. Data Collection and Calculation*

In this paper, the original data included are mainly obtained through the authors' rural survey. It is difficult to count agricultural input–output data, and the intermediate input between products are also complex. To ensure the accuracy of the data, 41 villages in Zoumajie town were followed up for one year. The agricultural input–output original data of Zoumajie town in 2018 were obtained by using the average agricultural input–output data of each village. The research results of H.T. Odum and Lan Shengfang were taken as the main references for the application of the emergy conversion rate in this paper [14–16]. The energy conversion coefficient of various agricultural production factors mainly referred to the Handbook of Agricultural Technology and Economy in China and the research results of Zhu Yulin [17]. Then, the input–output data of the agro-ecological economic system involved in this paper were processed.

*3.3. Research Results*

3.3.1. The Basic Situation of the Emergy Input and Output

Based on the table of emergy input–output of the agricultural eco-economic system in Zoumajie town (Table 3) in 2018, the total economic emergy output of the town's planting industry products was $4.25 \times 10^{20}$ sej. The total output of its animal husbandry products was $7.11 \times 10^{19}$ sej. The total output of its forestry products was $2.88 \times 10^{19}$ sej. The total output of its fishery products was $2.73 \times 10^{19}$ sej. The total economic emergy of agricultural and sideline was $2.30 \times 10^{20}$ sej. Among them, the only agricultural and sideline product in this area is straw, which is mainly used for agricultural production activities and does not directly generate economic value. Therefore, no comparison is made. In the agricultural ecosystem of the town, the economic value of planting products is the largest, followed by that of animal husbandry products. The economic value of forestry products ranks third, and fishery products have the smallest economic value. Therefore, in 2018, the sources of economic benefits in the agricultural ecosystem of the town were planting products and animal husbandry products.

In this table, each row of data represents the economic emergy output of the corresponding sector and the flow of economic emergy invested in agricultural activities. The data of the first row show that the economic emergy of planting products input into planting products in the form of seeds or tubers is $1.16 \times 10^{17}$ sej. The economic emergy of husbandry products and fishery products in the form of feed is $1.81 \times 10^{19}$ sej and $1.34 \times 10^{18}$ sej, respectively. Planting products have no investment in forestry products. The economic emergy of planting products reserved for self-sufficiency consumption is $7.58 \times 10^{19}$ sej, which will not increase inventory. The economic emergy of planting products for sale is $7.02 \times 10^{19}$ sej. Planting products with an economic emergy of $3.05 \times 10^{19}$ sej are sent for processing. The economic emergy of planting products is $-1.04 \times 10^{18}$ sej in the balance difference. This result is a negative number, which means when planting products are produced, some products' input into agricultural production activities are imported from the outside world and are not included in the total economic emergy output. According to the survey, the seeds needed for planting agricultural products are all purchased in this area. Every year, agricultural activities do not involve storing seeds for planting in the coming year. When planting products are used as feed for animal husbandry products and fishery products, some of them are also obtained through purchase.

**Table 3.** Emergy input–output table of the agro-ecological economic system in Zoumajie town.

| | | Intermediate Use | | | | | End Use | | | | Total Final Use | Balanced Differences | Total Output |
| | | | | | | | Agricultural Self-Retention | | Commodity Products | | | | |
| | | Planting Industry | Animal Husbandry | Forestry | Fishery | Agricultural and Sideline | Self-Sufficiency Consumption | Storage | Non-Agricultural Intermediate Products | Final Agricultural Products | | | |
|---|---|---|---|---|---|---|---|---|---|---|---|---|---|
| Agriculture | Planting industry | $1.16 \times 10^{17}$ | $1.81 \times 10^{19}$ | 0 | $1.34 \times 10^{18}$ | $2.30 \times 10^{20}$ | $7.58 \times 10^{19}$ | 0 | $3.05 \times 10^{19}$ | $7.02 \times 10^{19}$ | $1.77 \times 10^{20}$ | $-1.04 \times 10^{18}$ | $4.25 \times 10^{20}$ |
| | Animal husbandry | 0 | $9.76 \times 10^{18}$ | 0 | 0 | 0 | $2.24 \times 10^{19}$ | 0 | 0 | $4.18 \times 10^{19}$ | $6.42 \times 10^{19}$ | $-2.89 \times 10^{18}$ | $7.11 \times 10^{19}$ |
| | Forestry | $3.89 \times 10^{17}$ | 0 | $2.36 \times 10^{17}$ | 0 | 0 | $2.11 \times 10^{18}$ | $6.48 \times 10^{18}$ | $5.25 \times 10^{18}$ | $8.16 \times 10^{18}$ | $2.20 \times 10^{19}$ | $6.14 \times 10^{18}$ | $2.88 \times 10^{19}$ |
| | Fishery | 0 | 0 | 0 | $2.11 \times 10^{18}$ | 0 | $3.43 \times 10^{18}$ | 0 | 0 | $1.45 \times 10^{19}$ | $1.79 \times 10^{19}$ | $7.22 \times 10^{18}$ | $2.73 \times 10^{19}$ |
| | Agricultural and sideline | $1.78 \times 10^{20}$ | $1.57 \times 10^{19}$ | 0 | $3.63 \times 10^{18}$ | 0 | $1.69 \times 10^{19}$ | 0 | 0 | 0 | $1.69 \times 10^{19}$ | $1.58 \times 10^{19}$ | $2.30 \times 10^{20}$ |
| Industrial auxiliary energy | Fertilizer | $4.58 \times 10^{19}$ | 0 | $8.93 \times 10^{18}$ | 0 | 0 | | | | | | | |
| | Pesticide | $1.24 \times 10^{17}$ | 0 | $4.49 \times 10^{16}$ | 0 | 0 | | | | | | | |
| | Agricultural film | $6.37 \times 10^{11}$ | 0 | 0 | 0 | 0 | | | | | | | |
| | Agricultural machinery | $6.21 \times 10^{17}$ | $8.84 \times 10^{14}$ | $1.71 \times 10^{16}$ | $1.25 \times 10^{17}$ | 0 | | | | | | | |
| | Agricultural diesel | $2.23 \times 10^{18}$ | 0 | 0 | 0 | 0 | | | | | | | |
| | Agricultural electricity | $4.78 \times 10^{17}$ | $1.42 \times 10^{17}$ | $9.55 \times 10^{16}$ | $2.99 \times 10^{17}$ | 0 | | | | | | | |
| | Concentrated feed | 0 | $1.47 \times 10^{19}$ | 0 | $1.05 \times 10^{18}$ | 0 | | | | | | | |
| Organic energy | Labour | $1.97 \times 10^{20}$ | $1.22 \times 10^{19}$ | $1.94 \times 10^{19}$ | $1.46 \times 10^{19}$ | 0 | | | | | | | |
| | Animal power | $3.89 \times 10^{17}$ | 0 | 0 | 0 | 0 | | | | | | | |
| | Organic Fertilizer | $1.80 \times 10^{17}$ | 0 | $1.42 \times 10^{16}$ | 0 | 0 | | | | | | | |
| | Grass | 0 | $5.15 \times 10^{17}$ | 0 | $4.12 \times 10^{18}$ | 0 | | | | | | | |
| Total input | | $4.25 \times 10^{20}$ | $7.11 \times 10^{19}$ | $2.88 \times 10^{19}$ | $2.73 \times 10^{19}$ | $2.30 \times 10^{20}$ | | | | | | | |

The data of the second row show that husbandry products have no economic emergy input to planting products, forestry products or fishery products. The economic emergy input of animal husbandry products to animal husbandry products is $9.76 \times 10^{18}$ sej. The economic emergy input of husbandry is $-2.89 \times 10^{18}$ sej in the balance difference. Through research, we see that the piglets and calves used in husbandry production are all purchased in this area. The economic emergy of animal husbandry products reserved for self-sufficiency consumption is $2.24 \times 10^{19}$ sej, and the inventory have no increase, which means that the livestock products of pigs, cattle and sheep raised in the area each year are sold out or consumed by the farmers themselves. The economic emergy of the livestock products used for sale is $4.18 \times 10^{19}$ sej. There are no livestock products sent for processing, which indicates that livestock products are mainly for sale in this area.

The data of the third row show that forestry products have economic emergy input only to forestry products, which is $2.36 \times 10^{17}$ sej. Mainly camellia seeds, nuts and wood planting consume a certain number of seedlings. The economic emergy of forestry products reserved for self-sufficiency consumption is $2.11 \times 10^{18}$ sej. The economic emergy of increasing inventory is $6.48 \times 10^{18}$ sej. The economic emergy of forestry products used for sale is $2.16 \times 10^{18}$ sej. The economic emergy of forestry products sent for processing is $5.25 \times 10^{18}$ sej. The economic emergy of forestry products is $6.14 \times 10^{18}$ sej in the balance difference.

The data of the fourth row show that fishery products have economic emergy input only to fishery products, which is $2.11 \times 10^{18}$ sej. The economic emergy of fishery products reserved for self-consumption is $3.43 \times 10^{18}$ sej, without increasing inventory. The economic emergy of fishery products for sale is $1.45 \times 10^{19}$ sej, without fishery products sent for processing.

The data of the fifth row show that the economic emergy of agricultural and sideline products input into the planting industry in the form of fertilizer is $1.78 \times 10^{20}$ sej. The economic emergy of input into husbandry products and fishery products in the form of feed are $1.57 \times 10^{19}$ sej and $3.63 \times 10^{18}$ sej, respectively. No by-products are used in the production of forestry products. The economic emergy of the by-products reserved for self-consumption is $1.69 \times 10^{19}$ sej, which has no increase inventory. By-products will not be sold or sent for processing. The economic emergy of by-products is $1.58 \times 10^{19}$ sej in the balance difference. This is due to the fact that byproducts are not fully used in agricultural activities, resulting in some waste. Since by-products are generated along with the main products when the main products are produced, each department has economic emergy input to it, and as a companion product, the economic emergy input of other departments to by-products is no longer listed. The economic emergy input of by-products to other departments is zero.

The economic emergy input of industrial auxiliary energy in agricultural production activities are as follows: Chemical fertilizer has economic emergy inputs only for planting and forestry, which are $4.58 \times 10^{19}$ sej and $8.93 \times 10^{18}$ sej, respectively. Pesticides also have economic emergy inputs only for planting and forestry, which are $1.24 \times 10^{17}$ sej and $4.49 \times 10^{16}$ sej, respectively. Only agricultural production requires agro-film and agro-diesel, and the economic emergy inputs are $6.37 \times 10^{11}$ sej and $2.23 \times 10^{18}$ sej. The economic emergy inputs of agricultural machinery to planting, husbandry, forestry and fishery are $6.21 \times 10^{17}$ sej, $8.84 \times 10^{14}$ sej, $1.71 \times 10^{16}$ sej and $1.25 \times 10^{17}$ sej. The economic emergy inputs of agricultural electricity to planting, husbandry, forestry, and fishery are $4.78 \times 10^{17}$ sej, $1.42 \times 10^{17}$ sej, $9.55 \times 10^{16}$ sej and $2.99 \times 10^{17}$ sej. Only husbandry and fishery need to use concentrated feed during production, and the economic emergy inputs of concentrated feed for them are $1.47 \times 10^{19}$ sej and $1.05 \times 10^{18}$ sej.

The economic emergy input of organic energy to agricultural production activities are as follows: Each sector of agricultural production requires human input, and the economic emergy input of labor for planting, husbandry, forestry and fishery are $1.97 \times 10^{20}$ sej, $1.22 \times 10^{19}$ sej, $1.94 \times 10^{19}$ sej and $1.46 \times 10^{19}$ sej, respectively. Only planting production requires animal power, and its economic emergy input is $3.89 \times 10^{17}$ sej. Planting

and forestry production require organic fertilizer, and their economic emergy input is $1.80 \times 10^{17}$ sej and $1.42 \times 10^{16}$ sej, respectively.

Grass in this area is used as organic energy in agricultural production activities, however, it is used as feed only in husbandry and fishery production. It is not input as green manure into planting products and forestry products, and its economic emergy inputs for animal husbandry products and fishery products are $5.15 \times 10^{17}$ sej and $4.12 \times 10^{18}$ sej.

3.3.2. Direct Consumption Coefficients of Various Sectors

Based on the direct consumption coefficient table of the agricultural ecosystem in Zoumajie town (Table 4), each column of data in the table represents the direct consumption coefficient of the economic emergy for each sector in each row by the sector corresponding to that column. It also reflects the degree of dependence of that column on the direct economic emergy input of each sector.

As shown by the direct consumption coefficient of economic emergy in the first column, planting products have the biggest direct coefficient for the economic emergy of labor, which is 0.4630, followed by 0.4186 for the economic emergy of agricultural and sideline products, and the direct consumption coefficient for the economic emergy of fertilizer ranks third at 0.1077. Among agricultural sectors, the dependence of the production activities of planting products on the direct input of the economic emergy of labor is the largest. The level of modernization of agricultural development is still low; the dependence on the direct input of the economic emergy of planting products themselves is small, at only 0.0003. Among the industrial auxiliary energy sectors, the production activities of planting products have the most dependence on the direct input of the economic emergy of fertilizer. The direct consumption coefficient for the economic emergy of agricultural film is the smallest, close to zero, which means that the amount of agricultural film used in the production process of planting products is very small and its impact is negligible. Additionally, it has a low direct consumption coefficient for the economic emergy of agricultural machinery, agricultural diesel fuel and agricultural electricity, which can reflect the low level of use of agricultural machinery in the production activities of planting products in the region. Among the sectors of organic energy, the production activities of planting products are the most dependent on the direct input of the economic emergy of labor. The reliance on the direct input of the economic emergy of organic fertilizer and animal power is small. Additionally, the direct consumption coefficient for the economic emergy of organic fertilizer is only 0.0004, which reflects that the amount of organic fertilizer used in the production activities of planting products in the region is quite low.

Based on the direct consumption coefficient of economic emergy in the second column, the biggest direct consumption coefficient of livestock products is 0.2547 for the economic emergy of planting, followed by 0.2209 for the economic emergy of agricultural and sideline products. For the economic emergy of concentrated feed, the direct consumption coefficient ranks third at 0.2068. For the economic emergy of labour, the direct consumption coefficient ranks fourth at 0.1710. Among all agricultural sectors, the production activities of livestock products are more dependent on the direct input of the economic emergy of agricultural products, with the greatest dependence on the direct input of the economic emergy of planting products and a heavy reliance on the direct input of the economic emergy of by-products. These results mean that livestock products are mainly fed with planting products and by-products in the production process. Additionally, there is a relatively large reliance on the direct input of the economic emergy of livestock products themselves. Among the various sectors of industrial auxiliary energy, the production activities of livestock products have direct consumption for the economic emergy of only agricultural electricity, concentrated feed and agricultural machinery, with the greatest reliance on the direct input of the economic emergy of concentrated feed. The direct consumption coefficient for the economic emergy of agricultural machinery is the smallest, close to zero, which means that little agricultural machinery is used in the production process of livestock products. Among the organic energy sectors, in the production activities of livestock products, there

is the greatest reliance on the direct input of the economic emergy of labor. The smallest reliance is on the direct input of the economic emergy of grass, and the production activities of livestock products rely mainly on human labor.

**Table 4.** Direct consumption coefficients of various sectors of the agricultural eco-economic system in Zoumajie town.

| Products | Planting Products | Livestock Products | Forestry Products | Fishery Products | Agricultural and Sideline Products |
|---|---|---|---|---|---|
| Planting products | 0.0003 | 0.2547 | 0.0000 | 0.0492 | 1.0000 |
| Livestock products | 0.0000 | 0.1373 | 0.0000 | 0.0000 | 0.0000 |
| Forestry products | 0.0009 | 0.0000 | 0.0082 | 0.0000 | 0.0000 |
| Fishery products | 0.0000 | 0.0000 | 0.0000 | 0.0774 | 0.0000 |
| Agricultural and sideline products | 0.4186 | 0.2209 | 0.0000 | 0.1332 | 0.0000 |
| Fertilizer | 0.1077 | 0.0000 | 0.3103 | 0.0000 | 0.0000 |
| Pesticide | 0.0003 | 0.0000 | 0.0016 | 0.0000 | 0.0000 |
| Agricultural film | 0.0000 | 0.0000 | 0.0000 | 0.0000 | 0.0000 |
| Agricultural machinery | 0.0015 | 0.0000 | 0.0006 | 0.0046 | 0.0000 |
| Agricultural diesel | 0.0052 | 0.0000 | 0.0000 | 0.0000 | 0.0000 |
| Agricultural electricity | 0.0011 | 0.0020 | 0.0033 | 0.0110 | 0.0000 |
| Concentrated feed | 0.0000 | 0.2068 | 0.0000 | 0.0385 | 0.0000 |
| Labour | 0.4630 | 0.1710 | 0.6756 | 0.5350 | 0.0000 |
| Animal power | 0.0009 | 0.0000 | 0.0000 | 0.0000 | 0.0000 |
| Organic fertilizer | 0.0004 | 0.0000 | 0.0005 | 0.0000 | 0.0000 |
| Grass | 0.0000 | 0.0072 | 0.0000 | 0.1512 | 0.0000 |

The direct consumption coefficient of economic emergy in the third column shows that the biggest direct consumption coefficient of forestry products is 0.6756 for the economic emergy of labor, followed by 0.3103 for the economic emergy of fertilizer. In addition, among agricultural sectors, the production activities of forestry products have direct inputs of economic emergy to forestry products themselves. However, their direct consumption coefficient of economic emergy is relatively small, at 0.0082. Among the industrial auxiliary energy sectors, the production activities of forestry products have a small direct consumption coefficient for agricultural machinery, agricultural electricity and pesticides. It reflects the fact that the production activities of forestry products in the region use less agricultural machinery and rely the most on the direct input of chemical fertilizer. Among the organic energy sectors, there is the greatest reliance on the direct input of the economic emergy of labor. The reliance on the direct input of the economic emergy of organic fertilizer is small, and the production of forestry products mainly relies on people for tending.

Based on the direct consumption coefficient of economic emergy in the fourth column, fishery products have the biggest direct consumption coefficient of 0.5350 for the economic emergy of labor, followed by 0.1512 for the economic emergy of grass and 0.1332 for the economic emergy of agricultural by-products. Among agricultural sectors, the production activities of fishery products are the most dependent on the direct inputs of the economic emergy of by-products, followed by a relatively large dependence on the direct input of planting products. Among the industrial auxiliary energy sectors, the production activities of forestry products have a direct consumption for the economic emergy of only agricultural machinery, agricultural electricity and concentrated feed. However, the coefficient is small, it reflects that the production activities of forestry products in the region use less agricultural machinery. Among the organic energy sectors, there is the greatest reliance on the direct input of the economic emergy of labor. There is a relatively large reliance on the direct input of the economic emergy of grass. The production of fishery products relies mainly on

human care, while grass is used as feed for fishery products, which has a relatively great impact on fishery production activities.

### 3.3.3. The Complete Consumption Coefficient of Each Sector

As shown in the table (Table 5) of the complete consumption coefficients of the agro-ecosystem in Zoumajie town, each column of data in the table indicates the coefficient of complete consumption of economic emergy for each sector in each row by the sector corresponding to that column. It also reflects the degree of dependence of that column on the economic emergy input of each sector.

**Table 5.** The complete consumption coefficient of each department of the agricultural eco-economic system in Zoumajie town.

| Products | Planting Products | Livestock Products | Forestry Products | Fishery Products | Agricultural and Sideline Products |
|---|---|---|---|---|---|
| Planting products | 0.0903 | 0.3505 | 0.0158 | 0.1261 | 0.0000 |
| Livestock products | 0.0462 | 0.6399 | 0.0085 | 0.0251 | 0.0000 |
| Forestry products | 0.0000 | 0.0000 | 0.0031 | 0.0000 | 0.0000 |
| Fishery products | 0.0000 | 0.0000 | 0.0000 | 0.2151 | 0.0000 |
| Agricultural and sideline products | 1.1071 | 0.7414 | 0.0371 | 0.6319 | 0.0000 |
| Fertilizer | 0.4980 | 0.1019 | 0.1637 | 0.0587 | 0.0000 |
| Pesticide | 0.0013 | 0.0002 | 0.0020 | 0.0002 | 0.0000 |
| Agricultural film | 0.0000 | 0.0000 | 0.0000 | 0.0000 | 0.0000 |
| Agricultural machinery | 0.0094 | 0.0015 | 0.0004 | 0.0218 | 0.0000 |
| Agricultural diesel | 0.0149 | 0.0049 | 0.0003 | 0.0013 | 0.0000 |
| Agricultural electricity | 0.0090 | 0.0093 | 0.0018 | 0.0663 | 0.0000 |
| Concentrated feed | 0.0776 | 0.3629 | 0.0188 | 0.2303 | 0.0000 |
| Labour | 1.5459 | 1.0735 | 0.6347 | 1.4790 | 0.0000 |
| Animal power | 0.0080 | 0.0007 | 0.0001 | 0.0002 | 0.0000 |
| Organic fertilizer | 0.0186 | 0.0045 | 0.0017 | 0.0005 | 0.0000 |
| Grass | 0.0025 | 0.0135 | 0.0005 | 0.5454 | 0.0000 |

A comparison with Table 4 shows that the forestry and fishery products remain economically viable inputs only to their own products. However, planting products, livestock products and by-products have different levels of full economic emergy inputs to planting products, livestock products, forestry products and fishery products. At the same time, planting products, livestock products, forestry products and fishery products have different levels of full economic emergy consumption for all industrial auxiliary energy sectors and organic energy sectors.

The complete consumption coefficient for economic emergy shows that all products are the most dependent on the complete input of the economic emergy of by-products. The complete consumption coefficient of planting products for the economic emergy of by-products is 1.1071, which shows that the economic emergy input of by-products has a great influence on the production of planting products. In the production of livestock products, the complete consumption coefficient is second only to the complete consumption coefficient for the economic emergy of by-products. This result shows that the economic emergy input of livestock products also has a great influence on the production of livestock products. For forestry product production, the largest is also the economic emergy input of by-products, while the economic emergy inputs of planting products, livestock products and forestry products have an impact on it, although the impact is small. The greatest impact on the production of fishery products is still the economic emergy input of by-products, followed by the economic emergy input of fishery products themselves. These

results mean that the amount of fish fry input also has a comparatively large impact on the production of fishery products.

The economic emergy complete consumption coefficient for each sector of industrial auxiliary energy shows that all sectors have different degrees of complete consumption for the economic emergy of planting products, animal husbandry products, forestry products and fishery products. Planting products have the largest complete consumption coefficient for the economic emergy of chemical fertilizer, reaching 0.4980. This indicates that the planting products have the greatest dependence on the input of chemical fertilizer in the production process, while the input of other sectors has a certain impact on its production, although the impact is small. The input of economic energy of agricultural film is the smallest, which can be almost ignored.

In the production of livestock products, the complete consumption coefficient for the economic emergy of concentrated feed is the largest, at 0.3629. This result proves that the economic emergy input of concentrated feed has the greatest impact on livestock products. The complete consumption for economic emergy of fertilizer is second only to concentrated feed, which proves that the economic emergy of fertilizer input has a comparatively large impact on the production of livestock products. The production of forestry products is most dependent on the input of the economic emergy of fertilizer. The economic emergy input of concentrated feed has the greatest impact on the production of fishery products, and its economic emergy consumption coefficient is 0.2303.

As shown by the complete consumption coefficient of economic emergy in all organic energy sectors, the economic emergy input of labor has the greatest impact on the production of all products, which indicates that the production activities of all agricultural products in the region are mainly performed by human labor. The impact of the economic emergy input of livestock, organic fertilizer and grass are very small for most products, and only fishery products have some dependence on the economic emergy input of grass in their production.

## 4. Research Conclusions and Discussion

Based on the Wassily W. Leontief input–output model, this paper constructs an emergy input–output model of agricultural eco-economic system based on emergy theory. It overcomes the limitations of the traditional input-output model. First, the conversion rate of ecological energy is used to solve the problem posed by the fact that the material input-output table cannot be directly combined due to the different measurement units of different substances. Second, the model transforms the object of research from a pure economic system to an eco-economic system by adding resources and environmental factors. Third, this paper solves the problem posed by the fact that the value input-output model is greatly affected by inflation.

From the research, it is known that the current agricultural factor inputs in the agro-ecosystem of Zoumajie town, Shuangfeng City, are as follows: the production of planting products relies mostly on labor input, followed by inputs of by-products and chemical fertilizer. The production of livestock products also relies mostly on labor input, followed by inputs of by-products, livestock products, planting products, concentrated feed and chemical fertilizer. The production of forestry products relies mainly on labor input and, to a certain extent, on fertilizer input. The production of fishery products still relies mostly on labor input, followed by the input of by-products, grasses, and concentrated feeds. It can be seen that the agricultural ecosystem in this region is still in the stage of traditional agriculture, and agricultural activities rely heavily on labor, fertilizer and refined feed, while the use of agricultural machinery is less, and the use of organic fertilizer is quite small.

To promote the sustainable development of agriculture in Zoumajie town, the following adjustments can be made: the production of all products in the agro-ecosystem of this region is heavily dependent on the input of agricultural by-products, although there is waste in the use of straw as a by-product. Thus, the use of straw should be rationalized, and the utilization rate of straw should be improved. To transform traditional agriculture

into ecological agriculture, we should adjust the ratio of industrial auxiliary energy and organic energy input, reduce the amount of chemical fertilizer, concentrated feed and labor energy input, and increase the amount of organic fertilizer and agricultural machinery input to improve the operational efficiency of the agricultural system. The characteristics of agriculture should also be developed. The production of straw in the planting industry in this region is large; thus, we can use straw as the medium for edible fungus cultivation. The emergy output of animal husbandry products is also large; thus we can develop animal husbandry-related products, such as black pig cultivation. The agricultural development in the area consumes much labor; thus, labor resources can be combined with the development of the service industry to develop leisure, tourism and experiential agriculture. The town should develop ecological agriculture. It should use inorganic fertilizers to grow crops, such as human and livestock manure, to reduce pesticide, fertilizer input, and reduce pollution to the environment. Through the deep processing of agricultural products, the establishment of various agricultural products processing plants, increase farmers' income.

In this paper, the emergy input–output model of the agro-ecological economic system in Zoumajie town still has the following shortcomings: (1) This region of town is an open economic, ecological, and environmental system. A great deal of circulation and exchange of materials, energy and information take place between this region and the outside world. We combine emergy conversion with the material agricultural input–output model, however, emergy conversion cannot be combined with the value-based agricultural input–output model. Although the main materials in the economy and society are converted, the input and output of relevant environmental elements are not involved as it is difficult to quantify. We just set up a balanced differences term in the output quadrant; (2) This paper takes H.T. Odum's research results as the main reference for the emergy conversion rate, however, the choice of the emergy conversion rate is from Zhu Yulin's Agricultural Eco-Efficiency Research Around Dongting Lake. Although the emergy conversion rate proposed by H.T. Odum and others can be applied to most emergy analyses and the excessively long cycle has caused certain changes in the conversion level of matter and energy, there is often a certain error. In 2016, although Professor Mark T. Brown and others proposed a new emergy benchmark [18], they did not propose a new emergy conversion rate. Due to the overly complicated calculation process of the emergy conversion rate and the restriction of our personal level, this article cannot explore the optimal emergy conversion rate material. Therefore, based on these shortcomings, the following prospects are proposed: The emergy analysis method is the best measure of environmental resources for economic development, and in the future, if scholars in China can conduct more research on the calculation of the emergy conversion rate, the emergy research method will be greatly improved. Furthermore, how to combine the emergy conversion method with the value-based input–output model will be the direction of our future research.

**Author Contributions:** Conceptualization, Y.Z. and M.L.; methodology, S.L.; software, S.L.; validation, H.W., J.W. and W.W.; formal analysis, Y.Z.; investigation, S.L.; resources, S.L.; data curation, M.L.; writing—original draft preparation, S.L.; writing—review and editing, M.L.; visualization, M.L.; supervision, S.L.; project administration, Y.Z.; funding acquisition, Y.Z. All authors have read and agreed to the published version of the manuscript.

**Funding:** This research was funded by the following funds, and the authors take full responsibility for this paper: Natural Science Foundation of Hunan Province, grant number 2020JJ4950; 2019 annual project of the 13th Five-Year Plan for the Development of Philosophy and Social Science of Guangzhou, grant number 2019GZGJ210.

**Institutional Review Board Statement:** Not applicable.

**Informed Consent Statement:** Not applicable.

**Data Availability Statement:** The original data of this paper are mainly obtained through the authors' rural survey. The research results of H.T. Odum and Lan Shengfang were taken as the main references for the application of the emergy conversion rate in this paper. The energy conversion coefficient of

various agricultural production factors mainly referred to the Handbook of Agricultural Technology and Economy in China and the research results of Zhu Yulin.

**Conflicts of Interest:** The authors declare no conflict of interest.

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
