# Peer review of "Research on the Input–Output Model of the Rural Agricultural Eco-Economic System Based on Emergy Theory"

_sustainability, doi:10.3390/su14073717_

Round 1
Reviewer 1 Report
See the attached file.

Author Response
Thanks so much for your comments. We have supplemented the paper with some of the latest research literature which are important for the field of the paper.
Reviewer 2 Report
This paper does make three improvements to the original input-output table. Given that all regions are currently an open economic system, ecological system, and environmental system, the quadrant of environmental output related to the outside world in the input-output table is very important. The reviewers suggested two revisions. Please explain the following two points in the revised cover letter. First, please explain how the positive and negative benefits of the town's ecological environment spill over to the outside of the town are determined. Second, please add an explanation of how the positive and negative environmental elements input to the town from the outside are determined. If both of the above points are ignored, the limitations of the method should be explained in the conclusion section.
Author Response
Thank you so much for your comments. We combine emergy conversion with the physical agricultural input–output model, but emergy conversion cannot be combined with the value-based agricultural input–output model. Although the main materials in the economy and society are converted, the input and output of relevant environmental elements are not indeed involved, because it is difficult to quantify. The limitations of the method have been explained in the conclusion section according to your suggestion.
Reviewer 3 Report
I suggest the authors to modify and improve the paper in the following aspects.
- The paper reads too domain-specific. Even energy science researchers may feel difficult in reading and understanding the paper thoroughly. Please make the paper more understandable to almost all social scientists.
- Related to the first comment, the paper could suggest more general implications, especially policy implications.
- The paper reads lengthy. With clear hypotheses, the paper could cut down redundant and/or unnecessary explanation.
Author Response
Thank you so much for your suggestion and comments. Because the paper involves energy theory and input-output theory, the content of the paper is indeed difficult for non-specialized researchers to understand thoroughly. We try to make the article easy to understand. We have added some relevant policy recommendations in the conclusion of the paper. We have cut down redundant and/or unnecessary explanation, section 2.1, section 3.2 and other parts are involved.
Round 2
Reviewer 3 Report
I don't understand what was substantially modified with reflection of review comments. There are many typos and the style of in-text citations needs to be corrected.
Author Response
Thank you so much for your comments. We have corrected some grammatical and spelling mistakes in the article. We have revised some of the expressions in the article to make the paper more understandable. In addition, in the conclusion and discussion section of the paper, we have added some relevant policy suggestions. We have cut down some redundant and/or unnecessary explanation, Section 2.1, 2.2, 3.2 and other parts are involved. All revisions according to your comments have been marked up using the “Track Changes” function.
Round 3
Reviewer 3 Report
This is too a domain-specific paper, so I don't find any general implication for the audience. My second-round decision was already "reject."
Author Response
Dear Reviewer:
Thank you so much for your comments concerning our manuscript entitled “Research on the Input–Output Model of the Rural Agricultural Eco-Economic System Based on Emergy Theory” (ID: 1588385) again. Your comments are all valuable and very helpful for revising and improving our paper, as well as the important guiding significance to our researches.
We have studied the comments carefully and have made correction which we hope it is possible to meet your approval. The main corrections in the paper and the responds to your comments are as flowing:
Ecological economics is an interdisciplinary subject; this paper combines emergy theory with input-output analysis. In addition, some mathematical methods are applied in the process of input-output analysis. These factors lead to the paper being a domain-specific one, so it is difficult to understand for some sociologists. We have made revision of the paper again to improve the manuscript. We have revised and improved the contents of Sections 2.2.1, 2.2.2, 2.2.3, 2.2.5, 2.2.6, 3.2, etc.
What's more, these changes will not influence the content and framework of the paper. We appreciate your warm work earnestly, and hope that the correction will meet with approval.
Once again, thank you very much for your comments and suggestions.
Best regards for you.
Yu-lin Zhu